TOPICAL REVIEW

# Recent optical approaches for anatomical and functional dissection of neuron–astrocyte circuitry

Yoshiki Hatashita  and Takafumi Inoue

*Department of Life Science and Medical Bioscience, Faculty of Science and Engineering, Waseda University, Tokyo, Japan*

Handling Editors: Laura Bennet & Valentina Mosienko

The peer review history is available in the Supporting Information section of this article (https://doi.org/10.1113/JP287485#support-information-section).

**Abstract figure legend** This review summarises novel optical approaches to deciphering structural and functional wiring diagrams of neuron–astrocyte circuits in the brain. There are three groups: the anatomical proximity assay, transsynaptic viral tracing and the functional connection assay. The FRET-based neuron–astrocyte proximity assay allows mapping of ∼10 nm proximate synapse positioning within astrocytic territory with genetically defined neuron–astrocyte pairs. Transsynaptic neuron-to-astrocyte virus tracing provides optical and genetic access to circuit-defined astrocyte populations. Optical biosensors can reveal spatiotemporal dynamics of neuron–astrocyte signalling in both intra- and extracellular spaces from the nanoscale perisynaptic process to the astrocyte population level, and activity-dependent cell tagging tools for calcium and cFos gain access to astrocyte populations associated with functional connections with neurons and behaviour. In addition to the conventional approach with nanoscale imaging, these tools will facilitate further neuron–astrocyte circuit studies.

The Inoue lab investigates neuron and glia physiology at the cellular and molecular levels. **Yoshiki Hatashita** received his PhD from Waseda University in 2023, where he studied spatiotemporal dynamics of ATP gliotransmission and intracellular calcium activity in astrocytes under the supervision of Prof. T. Inoue. After gaining additional research experience on neuron–astrocyte circuits in the Inoue lab, he currently works as a postdoctoral researcher in the Group for Circuit Mechanisms of Behaviour at the University of Bonn Medical Centre in Germany. **Takafumi Inoue** is a professor at Waseda University, Tokyo, Japan. He started his research carrier in molecular biology by cloning an NFIA gene in the late 1980s and has long devoted himself to electrophysiology and cellular imaging with brain slices and recently with the *in vivo* mouse brain.

The Journal of Physiology

**Abstract** Astrocytes, the most abundant glial cells in the brain, are wired into neural circuits through close contact with neuronal pre- and post-synapses, called tripartite synapses. The mutual communication between neurons and astrocytes is crucial for neural circuit dynamics and animal behaviour. Recent advancements in imaging, manipulation and transcriptomics in astrocytes have revealed that astrocytes exhibit spatiotemporally complex computations and represent circuit-specialised functions and molecular makeups. However, understanding the neuron–astrocyte circuitry by means of conventional anatomical methods is hindered due to technical limitations. In this review, we highlight recently developed optical, genetic and viral techniques that enable high-throughput identification of connected neuron–astrocyte pairs with circuit and genetic specificity. These approaches will accelerate anatomical and functional dissections of the neuron–astrocyte circuits in health and disease in future studies.

(Received 21 October 2024; accepted after revision 12 February 2025; first published online 27 February 2025)

**Corresponding author** T. Inoue: Department of Life Science and Medical Bioscience, Faculty of Science and Engineering, Waseda University, Tokyo, Japan. Email: inoue.t@waseda.jp

## Introduction

Conceptually, the neuron–glia connection network lies on top of the network of neurons. Astrocytes, the most abundant glial cells, make contact with synaptic connections with their peculiar subcellular structure, perisynaptic astrocyte processes, which compose tripartite synapse with the pre- and post-synaptic structures of neurons (Araque et al., 1999; Ventura & Harris, 1999). The nanoscopic-level proximity of astrocytes to neurons was first associated with a supportive role in the uptake and removal of excess neurotransmitter molecules from synaptic clefts, but currently, bidirectional communication between astrocytes and neurons at tripartite synapses is envisaged: astrocytes influence synaptic transmission by actively releasing gliotransmitters and regulate synaptic plasticity, circuit formation and remodelling, and animal behaviour (Araque et al., 2014; Lyon & Allen, 2022; Nagai et al., 2021). In neurological disorders, astrocytes exhibit morphological, transcriptional and functional alterations, and some synaptic and behavioural dysfunctions can be rescued by astrocyte manipulation (Endo et al., 2022; Yu, Nagai, Marti-Solano et al., 2020). Astrocytes are now regarded as vital partners of neuronal circuits, and the mutual communication between astrocytes and neurons is considered a critical element for understanding brain function.

Each astrocyte has hundreds of thousands of synapses within its territory of non-overlapping processes (Bushong et al., 2002). This unique structure leads to the idea that each astrocyte actively affects the activities of a tremendous number of synaptic connections of heterogeneous neuron populations as a regional coordinator (Halassa et al., 2007). Various approaches are taken to gain a comprehensive understanding of astrocyte-neuron communication. Full reconstruction of single astrocytic territories at nanoscale resolution was recently achieved using electron microscopy in fixed brain tissues (Aten et al., 2022). Moreover, super-resolution microscopy enables live imaging of astrocytic nanostructure processes and their calcium activity *in vivo* (Arizono et al., 2020). The wide variety of patterns in astrocytic intracellular calcium dynamics in response to neuronal activity implies spatiotemporally complex computations inside astrocytes (Cahill et al., 2024; for a review: Semyanov, 2019; Wang et al., 2019), and recent efforts in direct optical readouts of gliotransmission have indicated local control of tens to hundreds of synapses by each single astrocyte (de Ceglia et al., 2023; Hatashita, Wu et al., 2023). However, due to the complex astrocytic nanoarchitecture and technical limitations, it is still challenging to determine the functional neuron–astrocyte connectivity of all synapses within an astrocytic territory, along with their neuronal origins and nanoscopic contact patterns. To build the entire picture of the neuron–astrocyte network, novel approaches to link multi-scale interrogations of anatomical and functional connections, ranging from nanoscopic perisynaptic structures and synapse positioning within single astrocytic territories to local and mesoscale neuron–astrocyte wiring diagrams, are required (Fig. 1).

This review aims to highlight emerging optical, genetic and viral tools for studying synaptic pairs of neuron–astrocyte connections in the rodent brain. These tools are classified into three groups: the anatomical proximity assay, transsynaptic tracing and the functional connection assay. We describe the advantages and current limitations of these techniques and discuss future directions towards the dissection of neuron–astrocyte circuits.

## Anatomical proximity assay

Perisynaptic astrocyte processes have ultrathin structures around 10 nm in diameter and are located at distances of around 50 nm from proximate synapses (Octeau et al., 2018; Salmon et al., 2023). Nanoscale imaging techniques, such as electron microscopy and super-resolution microscopy, are robust in resolving the detailed anatomy of tripartite synapses, but their imaging volume is limited for large-scale circuit dissection. Alternatively, a fluorescent protein-based tool capturing nanoscopic astrocyte synapse connections provides an efficient approach to depict structure-based wiring diagrams of neuron–astrocyte circuits with conventional optical set-ups.

**High-resolution microscopy.** Nanoscale imaging is the conventional approach to study structural astrocyte synapse interaction. Among high-resolution microscopy, electron microscopy has the highest spatial resolution of less than 10 nm. Imaging volume has been expanding, and the latest work using serial block face scanning electron microscopy has achieved reconstruction of the entire territory of single astrocytes (Aten et al., 2022). A computer vision-based tool tailored for the astrocyte structure was also developed to quantitatively analyse the nanoarchitecture geometry, such as the shortest distance between synaptic cleft and astrocyte surface (Salmon et al., 2023). However, electron microscopy is limited to

fixed tissue, and the sample preparation process can alter the astrocyte nanostructure. As an alternative approach, advanced fluorescence microscopy beyond the diffraction limit, namely super-resolution microscopy, has expanded the study of astrocytic nanostructure in more intact brain samples, even *in vivo*, along with multicolour molecular labelling (Heller et al., 2020) or functional recording (Arizono et al., 2020). Expansion microscopy also yields nanoscale resolution with conventional fluorescence microscopy set-ups (Herde et al., 2020). These techniques are highly compatible with fluorescence-based tools, therefore serving as powerful methods to anatomically validate the robustness of emerging circuit mapping techniques. Simultaneously, further advancements in imaging volume and automated analysis tools are highly desired.

**Neuron–astrocyte proximity assay.** While the above high-resolution methods are also useful in a wide range of biological applications, the neuron–astrocyte proximity assay (NAPA) offers a simple but effective approach particularly for investigating neuron–astrocyte circuit organisation. It uses Förster resonance energy transfer (FRET; Periasamy et al., 2008) to detect close astrocyte–neuron contacts at distances less than ∼10 nm under conventional confocal microscopy (Fig. 2A; Octeau et al., 2018). In this approach, a FRET donor, green fluorescent protein (GFP) fused with the trans-membrane domain of platelet-derived growth factor receptor (PDFGR), and an acceptor, mCherry fused with the transmembrane domain of Neurexin-1, are expressed on the plasma membrane of astrocytes and at the presynaptic axon terminals, respectively. A FRET signal reports colocalisation of GFP and mCherry, and therefore indicates that the distance between astrocyte and presynapse is within 10 nm (50% FRET efficiency at ∼5 nm distance); and microscopic colocalisation of the two tags without a FRET signal indicates that the distance is longer than 10 nm but within the diffraction limit, that is, hundreds of nanometres. Without using super-resolution microscopy, NAPA allows the evaluation of locations and densities of nanoscale proximate contacts between genetically specified neuron–astrocyte pairs under fixed and live conditions.

Using NAPA, Octeau et al. (2018) examined circuit-dependent neuron–astrocyte proximity in the striatum, which receives inhibitory collateral inputs from local medium spiny neurons, dopaminergic inputs from the substantia nigra, and glutamatergic inputs from the cortex and the thalamus. The measured FRET efficiency for the local collateral inputs was significantly the highest among the four inputs, and that for the nigrostriatal projection was the lowest, indicating pathway dependence on physical interactions of astrocyte–neuron

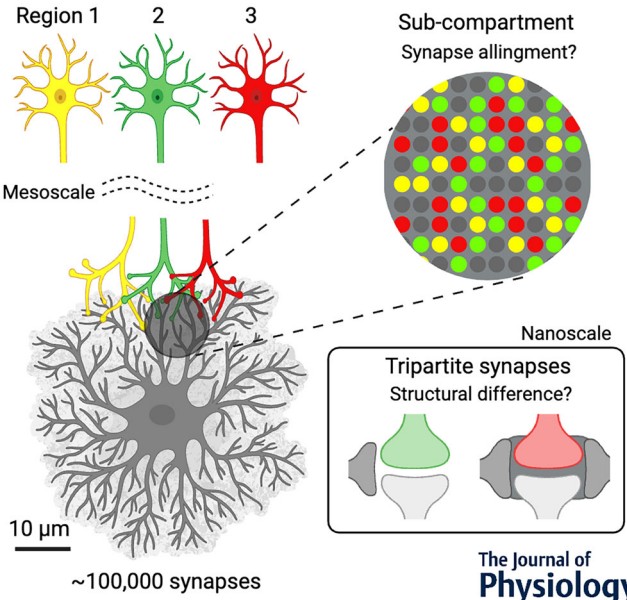

**Figure 1. Spatial organisation of neuron–astrocyte interactions**
A schematic illustration of the multiscale circuitry organisation of neuron-to-astrocyte projection. Pathway-dependent synapse positioning within an astrocyte territory and their nanoscopic tripartite synapse structures remain to be clarified.

connections at nanoscale. In a Huntington's disease mouse model, the FRET efficiency increased both in the corticostriatal and thalamostriatal pathways, but the FRET area decreased in the former but increased in the latter, suggesting circuit-specific alteration in the tripartite synapse structure in pathological conditions. Live imaging of NAPA in slices showed that the proximate contact less than 10 nm in the striatum was stable against electrical stimulation of afferents or under an ischaemic condition, oxygen–glucose deprivation (Octeau et al., 2018).

Potential applications of NAPA with further variations can be suggested. First, brain-wide FRET donner expression in the astrocyte plasma membrane with intravenous adeno-associated virus (AAV) serotype PHP.eB (AAV-PHP.eB) vector injection or using transgenic mice together with neuron type- or brain region-selective FRET acceptor expression would facilitate the assessment of region-dependent astrocyte proximity to neuronal projections. Second, additional FRET colour variations would enable simultaneous comparison of contacts with different neurons in single astrocyte territories. NAPA

with FRET pairs of longer Förster distances would also be favourable, given that the average distance between a glutamatergic synapse and an astrocyte is 50 nm (Octeau et al., 2018). Third, *in vivo* live NAPA imaging under two-photon microscopy will enable chronic tracking of the structural remodelling process of tripartite synapses under physiological and pathological conditions with circuit identities.

## Transsynaptic viral tracing

Genetic engineering and improvements in transneuronal viral tracers have greatly advanced anatomical and functional circuit mapping in the brain with cell type and circuit specificities. Transneuronal viral tracers commonly used include Herpes simplex virus (HSV), vesicular stomatitis virus (VSV) and AAV for anterograde tracing and rabies virus (RV) and pseudorabies virus (PRV) for retrograde tracing (Xu et al., 2020). Recently, AAV serotype 1 (AAV1) and a live attenuated yellow fever virus (YFV) have been demonstrated to spread from neurons to astrocytes in an anterograde manner (Georgiou et al.,

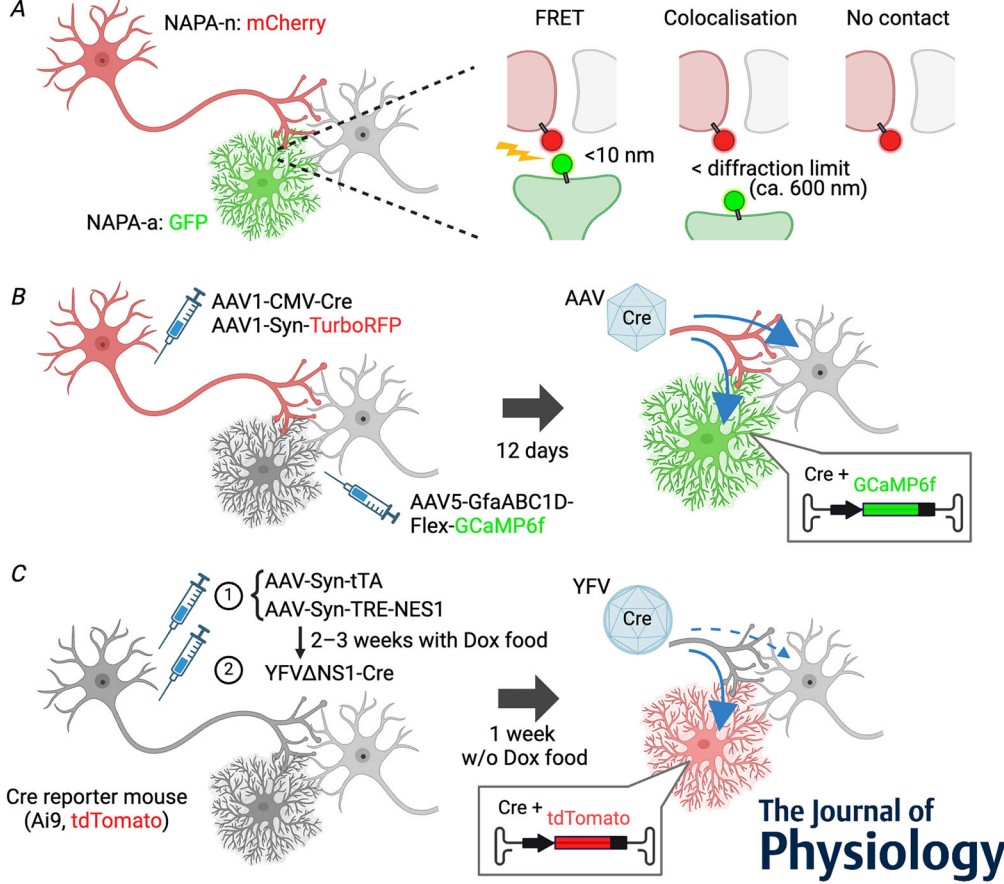

**Figure 2. Optical techniques for anatomical mapping of neuron–astrocyte connections**
Experimental designs of FRET-based proximity labelling using NAPA (*A*, Octeau et al., 2018) and transsynaptic anterograde tracing from neuron to astrocyte with AAV1-Cre (*B*, Georgiou et al., 2022) and with YFVΔNS1-Cre (*C*, Thompson et al., 2023).

2022; Thompson et al., 2023), opening the door to virally mapping the neuron–astrocyte circuits.

**AAV1-Cre.** AAVs are widely used for gene delivery in neuroscience thanks to their minimal cell toxicity, relatively stable gene expression, and various serotype choices with unique cell tropisms. AAV1 and AAV9 were found to cross neuronal synapses anterogradely. When these AAV vectors encode the Cre gene (AAV-Cre) and postsynaptic neurons are equipped with a Cre-dependent gene expression switch, anterograde transfer of AAV turns on the gene expression in the postsynaptic neurons (Zingg et al., 2017). AAV1-Cre is more commonly used as an anterograde transsynaptic tracer.

Georgiou et al. demonstrated anterograde tracing from neuron to astrocyte using the AAV1-Cre system for the first time in the thalamic ventral posterior medial nucleus (VPM) to barrel cortex pathway (Georgiou et al., 2022). Following a 12-day transduction period of AAV1-CMV-Cre in the VPM, not only neurons but also astrocytes in cortex contained AAV capsids in their cell bodies. Astrocyte-selective anterograde tracing (Fig. 2*B*) was achieved by adopting an AAV vector carrying a Cre-dependent gene expression cassette under control of an astrocyte-specific promoter fragment (GfaABC1D). Co-injection of AAV1-CMV-Cre and AAV1-hSyn-TurboRFP into the VPM and injection of AAV5-GfaABC1D-Flex-lck-GCaMP6f into the barrel cortex allowed simultaneous monitoring of axonal structures and innervated astrocytic calcium dynamics in the barrel cortex of awake mice. The thalamocortical axon–astrocyte tracing with AAV1-CMV-Cre was further confirmed by a follow-up study (Han et al., 2023). Hence, AAV1-Cre mediated tracing is a powerful approach allowing both anatomical and functional mapping between presynaptic neurons and astrocytes.

Neuron-to-astrocyte tracing with AAV1-Cre has opened new avenues to monitor and manipulate circuit-proper astrocyte populations. To date, however, whether the tracing is mediated through tripartite synapses or is due to the spillover of virus particles to extra-tripartite synapse space remains undetermined. It should be kept in mind that AAV1 also retrogradely crosses neuronal synapses, though less efficiently than anterograde transmission (Zingg et al., 2017, 2020). If this is also the case for neuron-to-astrocyte transfer, AAV1 tracing may not be ideal for intraregional local circuit analysis. In addition, AAV1 is transmitted to neurons from glutamatergic and GABAergic neurons but little or none is transmitted from cholinergic, noradrenergic and serotonergic neurons (Zingg et al., 2020). It is worth validating whether AAV1 has a neuron type-dependent bias in neuronal-to-astrocyte tracing.

**YFV.** YFV has recently been identified as an anterograde transneuronal virus and genetically modified for monosynaptic neuron-to-neuron (Li et al., 2021) and neuron-to-astrocyte tracing (Thompson et al., 2023). YFV is a positive-sense single-stranded RNA virus, and among the ten encoded proteins, non-structural protein 1 (NS1) is necessary for early RNA replication. In neuronal tracing, NS1-deleted YFV (YFV$\Delta$NS1) alone is incapable of fluorescence marker expression and transsynaptic propagation, but exogenous complementation of NS1 in both starter and postsynaptic neurons allows anterograde labelling. Retrograde propagation was observed late with 12 days latency, which was avoidable by temporal control of NS1 expression with the Tet-On system.

Native YFV has neurotropism: it predominantly infects neurons (Li et al., 2021). However, when YFV$\Delta$NS1-Cre infected neurons that were also complemented with NS1 in a Cre reporter transgenic mouse line, Ai9, in which cells turn on tdTomato expression by Cre recombinase activity, the virus selectively labelled astrocytes (93.7%) at the process terminals, even though NS1 was not expressed in astrocytes (Fig. 2*C*; Thompson et al., 2023). Such anterograde tracing of astrocytes from neurons was demonstrated in the motor, prefrontal, and lateral and medial entorhinal cortices, and the distributions of labelled astrocytes were in line with the neuronal projection patterns. The study also confirmed that the virus can be transmitted not only from glutamatergic neurons but also from GABAergic neurons. Thus, YFV$\Delta$NS1-Cre is a powerful tool for brain-wide neuron-to-astrocyte circuit mapping. However, YFV$\Delta$NS1-Cre did not label astrocytes when combined with an AAV vector carrying a Cre-dependent gene expression cassette instead of a Cre reporter transgenic mouse line, indicating room for improvement (Thompson et al., 2023). The study also found that YFV$\Delta$NS1-Cre directly infected astrocytes around the injected sites and turned on the reporter gene in astrocytes, which hampers intraregional tracing. Besides similarly to AAV1-Cre, the propagation mechanisms and functional and structural preferences of cell contacts are not characterised. It is particularly intriguing that YFV$\Delta$NS1-Cre anterogradely labels astrocytes but not neurons in Cre reporter mouse lines where all cells carry the Cre reporter. Thompson et al. speculated that astrocytes may collect viral particles more efficiently than neurons because of the dense interactions and larger surface area of the perisynaptic astrocyte structure, enough to turn on the Cre reporter gene expression even without NS1. Further clarification is needed on the preference of the virus for postsynaptic cell types and if the virus transfer is specifically mediated through the tripartite synapse.

## Functional connection assay

Astrocytes exhibit dynamic changes in second messenger levels and hence gene expression patterns in response to neuronal inputs through neurotransmitter receptors. To date, there are many options for capturing spatiotemporal dynamics of intra- and extracellular signalling from the subcellular domain to multicellular levels thanks to the great advance in genetically encoded fluorescent sensors (e.g. GCaMP series for intracellular calcium, intracellular cyclic AMP sensors: Kawata et al., 2022; Massengill et al., 2022; iGluSnFR for extracellular glutamate: Aggarwal et al., 2023; and GPCR-based sensors for extracellular neurotransmitters, neuromodulators and neuropeptides: Muir et al., 2024; also see: Hirrlinger & Nimmerjahn, 2022; Yu, Nagai et al., 2020). Several software packages have been developed to analyse the complex calcium dynamics in astrocytes that are also applicable for other optical probes (for a review: Hirrlinger & Nimmerjahn, 2022; Wang et al., 2019). In addition, activity-dependent cell tagging techniques provide optical and genetic access to astrocyte populations that are functionally linked to specific circuits or animal behaviours (Serra et al., 2022; Williamson et al., 2024).

**Calcium imaging.** Calcium imaging has long been used to measure the neuron-to-astrocyte interaction, as astrocytes respond to almost all types of neuro-transmitters and neuromodulators with intracellular calcium elevation. Two-photon calcium imaging allows subcellular functional mapping of astrocytic micro-domains responsive to neuronal inputs *in vivo* (Georgiou et al., 2022; Stobart, Ferrari, Barrett, Stobart et al., 2018), and fluorescence lifetime imaging revealed heterogeneous resting calcium levels at nanomolar range within and among astrocytes (Zheng et al., 2015). The calcium response patterns vary against different neurotransmitters and show local, propagative and global elevations, indicating spatiotemporally complex information processing (Cahill et al., 2024; for a review: Semyanov, 2019; Wang et al., 2019). In the hippocampus, the spatial position of an animal can be decoded from calcium responses in astrocyte populations, supporting the idea that the calcium activity of astrocytes encodes sensory information and mirrors neuronal activities well (Curreli et al., 2022; Doron et al., 2022). Recent applications in perisynaptic nanoscale calcium imaging with super-resolution microscopy (Arizono et al., 2020) and multicolour two-photon imaging of calcium activities in astrocytes and axons (Reynolds et al., 2019), neuro-transmitter dynamics (Oe et al., 2020) and gliotransmitter dynamics (Hatashita, Wu et al., 2023) have also promoted in-depth examination of functional communication between neuron and astrocyte *in vivo*.

Serra et al. (2022) conducted mesoscale functional mapping of pathway-dependent astrocytic calcium response in nucleus accumbens (NAc) using a genetically encoded photoconvertible fluorescent probe, calcium-modulated photoactivatable ratiometric integrator (CAMPARI), in brain slice. CAMPARI detects cells with high calcium activity histories by green-to-red photoconversion under violet light illumination together with real-time calcium dynamics. When glutamatergic projections from the medial prefrontal cortex (mPFC), amygdala and ventral hippocampus (vHip) were each optogenetically stimulated, the postsynaptic neuronal response patterns aligned with the respective stimulated axonal projections. By contrast, the activated astrocyte distributions only partially overlapped with the projection patterns, suggesting that functional neuron-to-astrocyte connections do not simply follow neuronal wiring patterns. Moreover, simultaneous activation of the mPFC and vHip afferents significantly amplified astrocytic calcium activities, while that of other combinations resulted in weaker responses than each single projection activation, showing pathway-dependent input integration. Pioneering works using organic calcium indicators and thin-tip glass electrodes for neuronal stimulation in brain slice preparation revealed that astrocytes responded in a projection-selective manner in hippocampus (Schaffer collaterals *vs.* alveus; Perea & Araque, 2005) and barrel cortex (inner *vs.* neighbour column; Schipke et al., 2008). Together, these studies suggest that astrocytes discriminate neuronal inputs in projection-dependent manners that are independent of the neuron-to-neuron connection properties.

Calcium imaging is the first option for functional examination of the neuron-to-astrocyte connection, but several potential caveats should be noted. First, functional activity mapping of astrocyte response by means of calcium activity may involve polysynaptic connections considering the slow kinetics of astrocytic response (but see Stobart, Ferrari, Barrett, Glück et al., 2018 for rapid calcium response). Using appropriate inhibitors for related receptors, locally blocking neuronal activity with tetrodotoxin, or lateral or feedforward inhibitions with picrotoxin may help with in-depth examinations, though it is still challenging to completely rule out the possibility of polysynaptic connections. Simultaneous imaging of astrocytic calcium dynamics with neuro-nal bouton calcium activities (Reynolds et al., 2019) or with neurotransmitter dynamics (Oe et al., 2020) would also help in examining the functional monosynaptic connection. Second, conventional single-plane imaging with two-photon microscopy covers 4% of the entire volume of single astrocytes, missing ∼90% of calcium events that volumetric two-photon calcium imaging would detect (Bindocci et al., 2017). Third, calcium imaging provides only partial aspects of functional

processes. Another potent second messenger, cyclic AMP, plays key roles in glycogen metabolism, structural regulation, and learning and memory in astrocytes (Reuschlein et al., 2019). Simultaneous imaging of calcium and cyclic AMP in astrocytes revealed their different temporal properties and that the latter requires sustained neuronal inputs (Oe et al., 2020). Towards a better understanding of the neuron–astrocyte circuit dynamics, it is highly encouraged to foster imaging of additional signalling molecules in astrocytes.

**cFos tagging.** Immediate-early genes, such as cFos, Arc and Npas4, are used as markers for active neurons associated with behaviour, such as learning and memory (Josselyn & Tonegawa, 2020). Several groups showed that the cFos expression increases not only in neurons but also in astrocytes following the activation of presynaptic neurons or astrocytic GPCR signalling *in vivo* (Adamsky et al., 2018; Nagai et al., 2019). More recently, Williamson et al. (2024) have developed an astrocyte-targeted cFos tagging system to gain genetic access to astrocyte populations associated with memory-related behaviour.

For the astrocytic cFos tagging, an AAV vector carrying a Cre-switching Flp recombinase cassette under a Fos promoter control (AAV-Fos-Flex-Flp) was injected to the hippocampus of a transgenic mouse line in which astrocytes express Cre under tamoxifen (Aldh1|1-CreER) crossed with an Flp reporter mouse line that expresses tdTomato after recombination by Flp (Rosa-CAG-FSF-tdTomato) (Williamson et al., 2024). Three days of tamoxifen administration followed by contextual fear conditioning resulted in tdTomato expression in a subset of hippocampal astrocytes. The tdTomato-positive astrocytes exhibited enhanced calcium activity in amplitude and duration, suggesting physiological alteration in some sets of astrocytes after fear learning. Combining with neuronal cFos tagging, the study demonstrated that the labelled astrocytes preferentially interact with engram neurons. Additionally, chemogenetic reactivation of cFos-tagged astrocytes but not in bulk was sufficient for fear memory retrieval. Moreover, transcriptomic analysis identified several differentially expressed genes specific to the tagged astrocytes, and the study further demonstrated that one of the genes, *NF1A* encoding a transcription factor, is crucial for memory performance in cFos-tagged astrocytes. It should be kept in mind that the number of cFos-tagged astrocytes depends on the dosage and schedule of tamoxifen administration. Besides, precise temporal control is limited as several days of consecutive tamoxifen administration is required. However, this approach potentially enables whole-brain mapping of astrocytes functionally linked to certain neuronal circuits or behaviours that would be challenging for the

CAMPARI system due to the strong light scattering in the living tissue.

## Outlook and future challenges

The emerging techniques mentioned above allow us to study the neuron–astrocyte connections with genetic and circuit specifications. Here, we highlight critical questions and technical challenges to be resolved in future studies to advance our understanding of the neuron–astrocyte circuits.

**Structural and molecular profiles of tripartite synapses and circuit specificity.** The finding that astrocytes respond in a pathway-specific manner (Martín et al., 2015; Perea & Araque, 2005; Schipke et al., 2008; Serra et al., 2022) provokes the question of how astrocytes discriminate between inputs. One possibility is that the perisynaptic contact structures differ in a circuit-specific manner. Around 40–60% of the synapses are touched by astrocytes at the cleft, and fractions of perimeter coverage by astrocytes and the perisynaptic astrocyte structure vary, indicating diverse contact patterns within each single astrocyte (Arizono et al., 2020; Aten et al., 2022; Lanjakornsiripan et al., 2018; Salmon et al., 2023; Ventura & Harris, 1999). Differences in the neuron–astrocyte proximity at tripartite synapses according to the presynaptic projection origin revealed by NAPA also support this possibility (Octeau et al., 2018).

Another possibility is that the molecular makeup of tripartite synapses that regulates functional and structural connectivity varies depending on the projection pathway. Several cell adhesion molecules (CAMs) important for neuronal connection are also enriched in perisynaptic astrocyte processes (for a review: Ngoc et al., 2024). In the hippocampus, the Eph and ephrin interaction regulates astrocytic glutamate transporter expression as well as spine density and synaptic morphology (Carmona et al., 2009). Transcriptome analysis suggested layer-specific cadherin expression in both neurons and astrocytes in the cortex, and N-cadherin (Cdh2) was shown to regulate astrocyte morphology specifically in deep cortical layers (Tan et al., 2023). HepaCAM expression on the astrocyte side of tripartite synapses was observed mainly at inhibitory connections and some at excitatory connections, preferring intracortical (∼20%) over thalamocortical (∼6%) projections in the cortical layer 1 (Baldwin et al., 2021), which implies the possibility of CAM-mediated circuit- or cell type-dependent interactions and structural and functional control of tripartite synapses. Recently, chemico-genetic proximity labelling with TurboID enabled proteome analysis specific to tripartite synapses (Takano et al., 2020). This technique can be used to characterise the pathway-specific proteome

profiles. Viral tracing and activity-dependent cell tagging systems are useful for detecting neuron–astrocyte pairs, whereas NAPA and super-resolution microscopy offer effective and robust anatomical approaches. Clarification of the functional relevance of the tripartite synapse with molecular and structural specificities is awaited.

**Spatial alignment of synapses within a territory of single astrocytes.** The next question is if and how synaptic distribution is spatially organised within individual astrocyte territories in a circuit-dependent manner. Single astrocytes are estimated to hold hundreds of thousands of synapses within their territories (Bushong et al., 2002). ATP released from cortical astrocytes spread over 50–250 $\mu m^2$ areas, suggesting that tens or even hundreds of synapses can be regulated in groups within each astrocytic territory (Hatashita, Wu et al., 2023). In hippocampal astrocytes, repeated calcium-dependent glutamate release was observed at particular spots, hotspots, which occupy ~30 $\mu m^2$, further suggesting specialised sub-compartments in the astrocytic structure (de Ceglia et al., 2023). For neuronal inputs, projection pathway-dependent synergistic amplification of astrocyte calcium response was reported, as mentioned above (Serra et al., 2022). Hence, the spatial alignment of synapses in the astrocytic arbour can be crucial in determining how astrocytes sort and integrate neuronal inputs and provide neuromodulation with or without target specificity. To resolve this, ideally all pre- and postsynaptic neuronal connections within the same astrocyte territory need to be identified simultaneously. While there are still large technical gaps, we propose that: (i) improvement of tools for simultaneous assay for multiple pathways, (ii) development of tools to capture connections between astrocytes and pre- and postsynaptic neurons, and (iii) transsynaptic tracing methods from astrocytes to neurons would help fill this gap. We recently adopted a retrograde transsynaptic tracer RV and showed virus transmission from astrocytes to presynaptic neurons (Hatashita, Li et al., 2023; also see astrocyte–microglia tracing: Clark et al., 2021). Further development and validation of viral tracers for neuron–astrocyte connections could be one of the promising next steps.

**Microcircuits unique to specific astrocyte subpopulation.** In contrast to the classical view that astrocytes are homogeneous cells, accumulating evidence demonstrates morphological, transcriptional and functional heterogeneity of astrocytes across and within brain regions (Bayraktar et al., 2020; Chai et al., 2017; Endo et al., 2022; Lanjakornsiripan et al., 2018). Recent studies indicate functionally specialised microcircuits involving specific astrocyte subpopulations. In the hippocampus, astrocytes can be classified into nine groups based on the transcriptional profiles, and one subpopulation termed 'glutamatergic astrocytes' is specialised to release glutamate (de Ceglia et al., 2023). This subpopulation is located across the hippocampus and most densely in the dentate gyrus, and astrocyte-specific knockdown of vesicular glutamate transporter 1 impaired contextual fear memory. In the central amygdala, the oxytocin receptor-expressing ($OT^+$) astrocyte subpopulation (~20%) is interspersed with and transmits oxytocin-evoked signalling to adjacent non-$OT^+$ astrocytes through gap junctions and to inhibitory neurons via D-serine release, consequently attenuating anxiety levels of mice (Wahis et al., 2021). In the cortex, astrocytes expressing chordin-like 1 (Chrdl1), an astrocyte-secreted synaptogenic factor, are enriched in the upper layer, and Chrdl1 regulates synapse maturation more preferentially in thalamocortical rather than intracortical connections (Blanco-Suarez et al., 2018). In the striatum, astrocytes expressing *Crym* are enriched in the central region with a high expression level of GABA transporters (Ollivier et al., 2024). *Crym* is associated with neurodegenerative disorders such as Huntington's disease, and its knockdown resulted in excessive repetitive behaviour and altered orbitofrontal corticostriatal signalling (Ollivier et al., 2024). All these accumulating lines of evidence suggest that the molecularly defined astrocyte subpopulations are linked to particular circuit alterations and behavioural dysfunctions, while if and how the molecularly and functionally defined astrocyte subpopulations have anatomically distinct wiring patterns with neurons remain unknown. Viral tracing and NAPA allow circuit dissection with cell type and brain regional specificity, thus being powerful tools to address this question. To facilitate this, further studies exploring subtype-selective astrocyte promoters or enhancers are also awaited.

## Concluding remarks

A quarter of a century has passed since the concept of the tripartite synapse was first introduced (Araque et al., 1999), and astrocytes are now regarded as a critical element of neuronal circuits. The finding that astrocytes distinguish among neuronal pathways and do not simply follow neuronal wiring diagrams warrants neuron–astrocyte circuit study. Recent advancements in astrocyte imaging with new sensors and manipulation with chemogenetics and optogenetics have significantly enhanced our understanding of astrocyte physiology and its involvement in synaptic transmission, circuit functions and animal behaviour. However, these interventions have been conducted on diverse astrocytes in bulk due to the lack of techniques to target specific astrocyte populations associated with circuits or functions, hindering clarification of the anatomical organisation of

neuron–astrocyte circuitry, inter- and intra-regional circuit diversity, and their functional and molecular relevance. This review has focused on emerging optical tools that enable high-throughput identification of connected neuron–astrocyte pairs and gain access to the astrocyte populations with circuit and genetic specificity. These tools are highly compatible with other genetically encoded sensors, tags and actuators and therefore will be the next breakthrough to accelerate future neuron–astrocyte research.

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

## Additional information

### Competing interests

The authors declare no conflict of interest.

### Author contributions

Both authors wrote the manuscript and approved the final version.

### Funding

The authors did not receive financial support for the submitted work.

### Author's present address

Y. Hatashita: Institute for Experimental Epileptology and Cognition Research, University of Bonn, Bonn, 53127, Germany.

### Keywords

neuron–glia interaction, proximity assay, transsynaptic tracing, tripartite synapse

### Supporting information

Additional supporting information can be found online in the Supporting Information section at the end of the HTML view of the article. Supporting information files available:

**Peer Review History**

