## [Peer Review History · The Journal of Physiology]

Recent optical approaches for anatomical and functional dissection of neuron-astrocyte circuitry

Yoshiki Hatashita and Takafumi Inoue
DOI: 10.1113/JP287485

Corresponding author(s): Takafumi Inoue (inoue.t@waseda.jp)

Review Timeline:

Submission Date:	21-Oct-2024
Editorial Decision:	11-Nov-2024
Revision Received:	05-Feb-2025
Accepted:	12-Feb-2025

Senior Editor: Laura Bennet

Reviewing Editor: Valentina Mosienko

Transaction Report:

Dear Dr Inoue,

Re: JP-TR-2024-287485 "Recent optical approaches for anatomical and functional dissection of neuron-astrocyte circuitry" by Yoshiki Hatashita and Takafumi Inoue

Thank you for submitting your manuscript to The Journal of Physiology. It has been assessed by a Reviewing Editor and by 2 expert referees and we are pleased to tell you that it is potentially acceptable for publication following satisfactory major revision.

Please address all the points raised and incorporate all requested revisions or explain in your Response to Referees why a change has not been made. We hope you will find the comments helpful and that you will be able to return your revised manuscript within 2 months. If your article is for a Special Issue, please note that we require your revised version within 2 months (rather than 9 months) in order to keep the Special issue on track. If you require longer than this, please contact journal staff: jp@physoc.org. Please note that this letter does not constitute a guarantee for acceptance of your revised manuscript.

ABSTRACT FIGURES: Authors are expected to use The Journal's premium BioRender account to create/redraw their Abstract Figures. Information on how to access this account is here:

<https://physoc.onlinelibrary.wiley.com/journal/14697793/biorender-access>.

REVISION CHECKLIST:

IMPORTANT POINTS TO NOTE WHEN REVISING YOUR MANUSCRIPT:

LANGUAGE EDITING AND SUPPORT FOR PUBLICATION: If you would like help with English language editing, or other article preparation support, Wiley Editing Services offers expert help, including English Language Editing, as well as translation, manuscript formatting, and figure formatting at www.wileyauthors.com/eoo/preparation. You can also find resources for Preparing Your Article for general guidance about writing and preparing your manuscript at www.wileyauthors.com/eoo/prepresources.

We look forward to receiving your revised submission.
If you have any queries, please reply to this email and we will be pleased to advise.

Yours sincerely,

Laura Bennet
Senior Editor
The Journal of Physiology

REQUIRED ITEMS

- Please include an Abstract Figure file, as well as the Figure Legend text within the main article file. The Abstract Figure is a piece of artwork designed to give readers an immediate understanding of the Review Article and should summarise the main conclusions. If possible, the image should be easily 'readable' from left to right or top to bottom. It should show the physiological relevance of the Review so readers can assess the importance and content of the article. Abstract Figures should not merely recapitulate other figures in the Review. Please try to keep the diagram as simple as possible and without superfluous information that may distract from the main conclusion of the Review. Abstract Figures must be provided by authors no later than the revised manuscript stage and should be uploaded as a separate file during online submission labelled as File Type 'Abstract Figure'. Please ensure that you include the figure legend in the main article file. All Abstract Figures will be sent to a professional illustrator for redrawing and you may be asked to approve the redrawn figure before your paper is accepted.

- Your MS must include a complete "Additional information section" with the following 4 headings and content:

Competing Interests: A statement regarding competing interests. If there are no competing interests, a statement to this effect must be included. All authors should disclose any conflict of interest in accordance with journal policy.

Author contributions: Each author should take responsibility for a particular section of the study and have contributed to writing the paper. Acquisition of funding, administrative support or the collection of data alone does not justify authorship; these contributions to the study should be listed in the Acknowledgements. Additional information such as 'X and Y have contributed equally to this work' may be added as a footnote on the title page.

It must be stated that all authors approved the final version of the manuscript and that all persons designated as authors qualify for authorship, and all those who qualify for authorship are listed.

Funding: Authors must indicate all sources of funding, including grant numbers. If authors have not received funding, this must be stated.

It is the responsibility of authors funded by RCUK to adhere to their policy regarding funding sources and underlying research material. The policy requires funding information to be included within the acknowledgement section of a paper. Guidance on how to acknowledge funding information is provided by the Research Information Network. The policy also requires all research papers, if applicable, to include a statement on how any underlying research materials, such as data, samples or models, can be accessed. However, the policy does not require that the data must be made open. If there are considered to be good or compelling reasons to protect access to the data, for example commercial confidentiality or legitimate sensitivities around data derived from potentially identifiable human participants, these should be included in the statement.

Acknowledgements: Acknowledgements should be the minimum consistent with courtesy. The wording of acknowledgements of scientific assistance or advice must have been seen and approved by the persons concerned. This section should not include details of funding.

- Please upload separate high quality figure files via the submission form.

- Author profile(s) must be uploaded via the submission form. Authors should submit a short biography (no more than 100 words for one author or 150 words in total for two authors) and a portrait photograph of the two leading authors on the

paper. These should be uploaded and clearly labelled together in a Word document with the revised version of the manuscript. Any standard image format for the photograph is acceptable, but the resolution should be at least 300 DPI and preferably more. A group photograph of all authors is also acceptable, providing the biography for the whole group does not exceed 150 words.

EDITOR COMMENTS

Reviewing Editor:

Excellent concise review on most of recent approaches to study astrocyte-neuron interaction/circuitry. The reviewers recommended discussing a few additional studies, approaches and tools which are relevant to the topic - please consider including those in the revised version of the manuscript.

REFEREE COMMENTS

Referee #1:

In this manuscript, Hatashita and Inoue discuss functional and physical astrocyte-neuron interactions and the optical tools available to study those interactions available to date.

I enjoyed reading the review but would encourage the authors to correct any grammatical errors to improve readability.

My main concern is that the authors fall short in describing why understanding neuron-astrocytes interactions is important, what these interactions could mean for brain function and why developing new optical tools would improve our understanding of brain (dys)function.

The authors don't discuss other intracellular second messengers that can and have been measured in astrocytes by e.g. FRET biosensors, including cAMP, or biosensor to measure extracellular neurotransmitters, such as iGluSnFR or ATP-sensitive probes. What other optical tools currently used to study neuronal physiology could be used to ultimately study astrocyte function?

Shortcoming of these techniques should be discussed as well. For example, the complexities that arise from the complex 3D morphology of astrocytes in vivo, and how this can affect interpretation of Ca²⁺ transients (e.g. PMID: 28522470).

The authors discuss the lack of tools to study circuit-specific astrocyte-neuron interactions, which should include limitations of current astrocyte-specific Cre systems and the lack of astrocyte subtype-specific transgenic animals.

If the authors want to discuss the molecular specialization of astrocytes to circuits or anatomical domains and what this means for brain function (as they seem to do starting on line 281), they will need to include the relevant literature, including for example PMID: 38418885 for Crym⁺ striatal astrocytes, PMID: 30344043 for Chrdl1⁺ L1 restricted astrocytes.

Finally, the Outlook and Concluding remarks section should again highlight why these optical tools are so important and what their future application could tell us about the astrocyte-neuron interaction and, critically, why it is so important to understand this interaction to understand circuit-function, brain function and behaviour.

Referee #2:

This is a timely and informative brief review which should be of interest to a rapidly growing pool of neuroscientists working in the field of astrocyte-neuron signaling. There are several points that the authors might want to address, to balance better their report in terms of the methods covered.

1. While "optical" is emphasized in the title, the review in large part describes genetic methods applied in combination with conventional optical techniques. There are, however, novel optical approaches that are clearly worth discussing in the present context, as briefly listed here:

Super-resolution STED imaging (PMID: 32312988) and diffraction-insensitive imaging methods (PMID: 32976770) have enabled monitoring live changes in perisynaptic astrocyte coverage, on the nanoscale. The expansion microscopy (PMID: 32966786) and single-molecule STORM (PMID: 32976770; PMID: 31153907) have begun to inform us about the astrocyte-associated 3D molecular nano-environment of synapses. A newly introduced FLIM-based method has allowed for intra-astrocyte Ca²⁺ monitoring in the nanomolar range (PMID: 32160550; PMID: 26494277) whereas STED imaging has provided nanoscopic resolution of Ca²⁺ dynamics in astrocyte processes (PMID: 32312988). A brief discussion of these advances should provide a more balanced overview, as far as novel optical methods are concerned.

2. Some of the described approaches (including the ones listed above) have certain limitations which are worth discussing. For instance:

- NAPA assay (FRET) should normally sense nanoscopic changes within the 5-10 nm range. It therefore remains uncertain how to interpret variation in FRET signals in terms of the neuronal-membrane adherence or the rearrangement of astrocyte processes, near or away from the actual synapses, given that the synaptic clefts are normally seen at >100 nm from the nearest astrocyte structure.

- Viral tracing of 'astrocyte-neuron connections' remains highly controversial, as partly explained by the authors. With no clear evidence for the corresponding synaptic or other junctional specializations, finding that an individual axon contacts a certain astrocyte has limited interpretational significance because the same axon crosses the territories of dozens or hundreds of astrocytes, and hundreds or thousands of axons cross the territory of the astrocyte under study.

Minor

Ref Hatashita et al (2023a) is an abstract. This is normally not acceptable for journal manuscripts, please consult the editors.

END OF COMMENTS

Response to the reviewers' comments

We thank the editor and reviewers for their time and efforts to improve the manuscript. According to their valuable comments, we revised our manuscript. Please find our below responses to specific issues raised by the reviewers. We believe that the changes in the manuscript and answers to the reviewers address all the issues raised by the reviewers.

EDITOR COMMENTS

Reviewing Editor:

Excellent concise review on most of recent approaches to study astrocyte-neuron interaction/circuitry. The reviewers recommended discussing a few additional studies, approaches and tools which are relevant to the topic - please consider including those in the revised version of the manuscript.

REFEREE COMMENTS

Referee #1:

In this manuscript, Hatashita and Inoue discuss functional and physical astrocyte-neuron interactions and the optical tools available to study those interactions available to date.

I enjoyed reading the review but would encourage the authors to correct any grammatical errors to improve readability.

(1) My main concern is that the authors fall short in describing why understanding neuron-astrocytes interactions is important, what these interactions could mean for brain function and why developing new optical tools would improve our understanding of brain (dys)function.

In accordance with the reviewer's suggestion, we have significantly revised the Introduction. Please see the revised manuscript with track changes for details (page 3, line 50). We believe this revision enhances the importance of the field and addresses the reviewer's concerns. We thank the reviewer for pointing out this issue.

(2) The authors don't discuss other intracellular second messengers that can and have been measured in astrocytes by e.g. FRET biosensors, including cAMP, or biosensor to measure extracellular neurotransmitters, such as iGluSnFR or ATP-sensitive probes. What other optical tools currently used

to study neuronal physiology could be used to ultimately study astrocyte function?

In line with the reviewer's comment, we have modified the manuscript as below. We thank the reviewer for their suggestions.

- We have specifically mentioned several optical biosensors currently available.

In the previous manuscript (page 6, line 187): “To date, there are many options for genetically encoded calcium sensors that exhibit better signal-to-noise ratios and optogenetics tools that allow selective neuronal stimulation.”

In the revised manuscript (page 8, line 240): “To date, there are many options for capturing spatiotemporal dynamics of intra- and extra-cellular signalling from the subcellular domain to multicellular levels thanks to the great advance in genetically encoded fluorescent sensors (e.g., GCaMP series for intracellular calcium, intracellular cyclic AMP sensors: Kawata *et al.*, 2022; Massengill *et al.*, 2022, iGluSnFR for extracellular glutamate: Aggarwal *et al.*, 2023, and GPCR-based sensors for extracellular neurotransmitters, neuromodulators, and neuropeptides: Muir *et al.*, 2024; also see: Yu *et al.*, 2020a; Hirrlinger & Nimmerjahn, 2022).”

- Regarding cAMP, we have also added a description below.

In the revised manuscript (page 10, line 306): “Third, calcium imaging provides only partial aspects of functional processes. Another potent second messenger, cyclic AMP, plays key roles in glycogen metabolism, structural regulation, and learning and memory in astrocytes (Reuschlein *et al.*, 2019). Simultaneous imaging of calcium and cyclic AMP in astrocytes revealed their different temporal properties and that the latter requires sustained neuronal inputs (Oe *et al.*, 2020). Towards a better understanding of the neuron-astrocyte circuit dynamics, it is highly encouraged to foster imaging of additional signalling molecules in astrocytes.”

- For further applications, we have added the description as:

In the revised manuscript (page 9, line 266): “Recent applications in perisynaptic nanoscale calcium imaging with super-resolution microscopy (Arizono *et al.*, 2020) and multicolour two-photon imaging of calcium activities in astrocytes and axons (Reynolds *et al.*, 2019), neurotransmitter dynamics (Oe *et al.*, 2020), and gliotransmitter dynamics (Hatashita *et al.*, 2023b) have also promoted in-depth examination of functional communication between neuron and astrocyte *in vivo*.”

(3) Shortcoming of these techniques should be discussed as well. For example, the complexities that

arise from the complex 3D morphology of astrocytes *in vivo*, and how this can affect interpretation of Ca²⁺ transients (e.g. PMID: 28522470).

Thank you for the careful review and insightful comment on the calcium imaging techniques. We have added a description as below.

In the revised manuscript (page 10, line 303): “Second, conventional single-plane imaging with two-photon microscopy covers 4% of the entire volume of single astrocytes, missing ~90% of calcium events that volumetric two-photon calcium imaging would detect (Bindocci *et al.*, 2017).”

(4) The authors discuss the lack of tools to study circuit-specific astrocyte-neuron interactions, which should include limitations of current astrocyte-specific Cre systems and the lack of astrocyte subtype-specific transgenic animals.

We agree with the comment that the lack of astrocyte subtype-specific promoters is one of the critical limitations. We have added the description below.

- In the revised manuscript (page 14, line 434): “To facilitate this, further studies exploring subtype-selective astrocyte promoters or enhancers are also awaited.”

(5) If the authors want to discuss the molecular specialization of astrocytes to circuits or anatomical domains and what this means for brain function (as they seem to do starting on line 281), they will need to include the relevant literature, including for example PMID: 38418885 for *Crym*⁺ striatal astrocytes, PMID: 30344043 for *Chrdl1*+ L1 restricted astrocytes.

We appreciate the reviewer’s suggestion regarding the related papers. The revised paper has included them as follows.

- (page 14, line 420): “In the cortex, astrocytes expressing Chordin-like 1 (*Chrdl1*), an astrocyte-secreted synaptogenic factor, are enriched in the upper layer, and *Chrdl1* regulates synapse maturation more preferentially in thalamocortical rather than intracortical connections (Blanco-Suarez *et al.*, 2018). In the striatum, astrocytes expressing *Crym* are enriched in the central region with a high expression level of GABA transporters (Ollivier *et al.*, 2024). *Crym* is associated with neurodegenerative disorders such as Huntington’s disease, and its knockdown resulted in excessive repetitive behaviour and altered orbitofrontal corticostriatal signalling (Ollivier *et al.*, 2024).

(6) Finally, the Outlook and Concluding remarks section should again highlight why these optical tools

are so important and what their future application could tell us about the astrocyte-neuron interaction and, critically, why it is so important to understand this interaction to understand circuit-function, brain function and behaviour.

Thanks to the reviewer's suggestion, we added the following modifications to highlight the significance of these tools.

- In the Outlook and future challenges section, we have added a description (page 12, line 377): "Viral tracing and activity-dependent cell tagging systems are useful for detecting neuron-astrocyte pairs, whereas NAPA and super-resolution microscopy offer effective and robust anatomical approaches."
- In the Concluding Remarks section, we have modified the description as:

Previous manuscript (page 10, line 303): "Recent advancements in astrocyte imaging with new sensors, manipulation with chemogenetics and optogenetics, and transcriptomics have significantly enhanced our understanding of astrocyte physiology and its involvement in synaptic transmission, circuit functions, and animal behaviour. However, the anatomical organisation of neuron-astrocyte circuitry, inter- and intra-regional circuit diversities, and their functional and molecular relevance remain largely mysterious. This review has focused on emerging tools that enable high-throughput identification of connected neuron-astrocyte pairs with circuit and genetic specificity. The tools are highly compatible with other genetically encoded sensors, tags, and actuators, therefore will be the next breakthrough to accelerate future neuron-astrocyte research.

Revised manuscript (page 14, line 439; added sentences in red): "**The finding that astrocytes distinguish among neuronal pathways and do not simply follow neuronal wiring diagrams warrants the neuron-astrocyte circuit study.** Recent advancements in astrocyte imaging with new sensors and manipulation with chemogenetics and optogenetics have significantly enhanced our understanding of astrocyte physiology and its involvement in synaptic transmission, circuit functions, and animal behaviour. However, **these interventions have been conducted on diverse astrocytes in bulk due to the lack of techniques to target specific astrocyte populations associated with circuits or functions, hindering clarification of** the anatomical organisation of neuron-astrocyte circuitry, inter- and intra-regional circuit diversities, and their functional and molecular relevance". This review has focused on emerging **optical** tools that enable high-throughput identification of connected neuron-astrocyte pairs **and gain access to the**

astrocyte populations with circuit and genetic specificity. These tools are highly compatible with other genetically encoded sensors, tags, and actuators and therefore will be the next breakthrough to accelerate future neuron-astrocyte research.

Referee #2:

This is a timely and informative brief review which should be of interest to a rapidly growing pool of neuroscientists working in the field of astrocyte-neuron signaling. There are several points that the authors might want to address, to balance better their report in terms of the methods covered.

(1) While "optical" is emphasized in the title, the review in large part describes genetic methods applied in combination with conventional optical techniques. There are, however, novel optical approaches that are clearly worth discussing in the present context, as briefly listed here:

Super-resolution STED imaging (PMID: 32312988) and diffraction-insensitive imaging methods (PMID: 32976770) have enabled monitoring live changes in perisynaptic astrocyte coverage, on the nanoscale. The expansion microscopy (PMID: 32966786) and single-molecule STORM (PMID: 32976770; PMID: 31153907) have begun to inform us about the astrocyte-associated 3D molecular nano-environment of synapses. A newly introduced FLIM-based method has allowed for intra-astrocyte Ca²⁺ monitoring in the nanomolar range (PMID: 32160550; PMID: 26494277) whereas STED imaging has provided nanoscopic resolution of Ca²⁺ dynamics in astrocyte processes (PMID: 32312988). A brief discussion of these advances should provide a more balanced overview, as far as novel optical methods are concerned.

We have revised the manuscript to reflect the suggestion by inserting a brief overview of the "Anatomical proximity assay" section and a new subsection titled "High-resolution microscopy", where we included expansion and super-resolution microscopy techniques (PMID: 32312988, PMID: 32966786, PMID: 31153907). Related to this, we have also modified the beginning of the "Neuron-astrocyte proximity assay" section.

- As the overview of the Anatomical proximity assay section (page 4, line 88)

“Perisynaptic astrocyte processes have ultrathin structures around 10 nm in diameter and are located at distances around 50 nm from proximate synapses (Octeau *et al.*, 2018; Salmon *et al.*, 2023). Nanoscale imaging techniques, such as electron microscopy and super-resolution microscopy, are robust in resolving the detailed anatomy of tripartite synapses, but their imaging volume is limited for large-scale circuit dissection. Alternatively, a fluorescent protein-based tool capturing nanoscopic astrocyte-synapse connection provides a simple approach to depict structure-based wiring diagrams of neuron-astrocyte circuits with conventional optical setups.

- The new subsection titled High-resolution microscopy (page 4, line 97):

“Nanoscale imaging is the conventional approach to study structural astrocyte-synapse interaction. Among the high-resolution microscopy, electron microscopy has the highest spatial resolution of less than 10 nm. Imaging volume has been expanding, and the latest work using serial block face scanning electron microscopy achieved reconstruction of the entire territory of single astrocytes (Aten *et al.*, 2022). A computer vision-based tool tailored for the astrocyte structure was also developed to quantitatively analyse the nanoarchitecture geometry, such as the shortest distance between synaptic cleft and astrocyte surface (Salmon *et al.*, 2023). However, electron microscopy is limited to fixed tissue, and the sample preparation process can alter the astrocyte nanostructure. As an alternative approach, advanced fluorescent microscopy beyond the diffraction limit, namely super-resolution microscopy, has expanded the study of astrocytic nanostructure in more intact brain samples, even *in vivo*, along with multicolour molecular labelling (Heller *et al.*, 2020) or functional recording (Arizono *et al.*, 2020). Expansion microscopy also yields nanoscale resolution with conventional fluorescent microscope setups (Herde *et al.*, 2020). These techniques are highly compatible with fluorescence-based tools, therefore serving as powerful methods to anatomically validate the robustness of emerging circuit mapping techniques. Simultaneously, further advancements in imaging volume and automated analysis tools are highly desired.”

- “Neuron-astrocyte proximity assay” section started with (page 5, line 117):

“While the above high-resolution methods are also useful in a wide range of biological applications, the neuron-astrocyte proximity assay (NAPA) offers a simple but effective approach particularly for investigating neuron-astrocyte circuit organisation.”

- In the Calcium imaging subsection, we mentioned the FLIM-based technique (page 9, line 258):

“...and fluorescent lifetime imaging revealed heterogeneous resting calcium levels at nanomolar range within and among astrocytes (Zheng *et al.*, 2015).”

(2) Some of the described approaches (including the ones listed above) have certain limitations which are worth discussing. For instance:

- NAPA assay (FRET) should normally sense nanoscopic changes within the 5-10 nm range. It therefore remains uncertain how to interpret variation in FRET signals in terms of the neuronal-membrane adherence or the rearrangement of astrocyte processes, near or away from the actual synapses, given that the synaptic clefts are normally seen at >100 nm from the nearest astrocyte structure.

- Viral tracing of 'astrocyte-neuron connections' remains highly controversial, as partly explained by the authors. With no clear evidence for the corresponding synaptic or other junctional specializations, finding that an individual axon contacts a certain astrocyte has limited interpretational significance because the same axon crosses the territories of dozens or hundreds of astrocytes, and hundreds or thousands of axons cross the territory of the astrocyte under study.

In line with the reviewer's comment on the limitation of NAPA, we have revised the manuscript as below. We are grateful for this comment.

- We have mentioned a potential application addressing the raised concern (page 6, line 154):
“NAPA with FRET pairs of longer Förster distances would also be favourable, given that the average distance between glutamatergic synapse and astrocyte is 50 nm (Octeau *et al.*, 2018).”
- We have modified the following description to clarify the limitation.
In the previous manuscript (page 3, line 85): “By live imaging, NAPA showed that **the contact proximity in the striatum was stable** against electrical stimulation of afferents and under an ischemic condition, oxygen-glucose deprivation (Octeau *et al.*, 2018).”
In the revised manuscript (page 5, line 143): “Live imaging of NAPA in slice showed that **the proximate contact less than 10 nm in the striatum was stable** against electrical stimulation of afferents or under an ischemic condition, oxygen-glucose deprivation (Octeau *et al.*, 2018).”

We appreciate the reviewer's concern regarding viral tracing. However, we believe our original content effectively addresses these limitations and aims to promote further validation of the viral tools.

Minor

Ref Hatashita et al (2023a) is an abstract. This is normally not acceptable for journal manuscripts, please consult the editors.

The paper posted on the SSRN preprint repository via Cell Press Sneak Peek follows a short paper style comprising Abstract, Results, and Discussion sections and four figures. The PDF can be obtained from the SSRN website (<https://dx.doi.org/10.2139/ssrn.4592585>) or Google Scholar (https://scholar.google.co.jp/citations?view_op=view_citation&hl=ja&user=sy2L9_0AAAAJ&citation_for_view=sy2L9_0AAAAJ:qjMakFHDy7sC). We noticed that the access link in the reference

section could be misleading (<https://www.ssrn.com/abstract=4592585>), therefore we have replaced it with “<https://dx.doi.org/10.2139/ssrn.4592585>”. We thank the reviewer for pointing out the concern.

Other changes

- We have added the **Abstract Figure** and **Abstract Figure Legend** (page 2).
- We have added the **Funding** and **Author profiles** sections (page 15, line 459)
- As a new functional approach, the study demonstrating astrocyte cFos tagging was published after our first submission. The following changes were made in the **Functional connection assay** section to include this technique in the revised manuscript.

The previous content is moved to the new subsection, **Calcium imaging** (page 9, line 253), and is followed by the **cFos tagging** subsection (page 11, line 314). For details, please see the revised manuscript file.

We added the overview at the start of the **Functional connection assay** section as (page 8, line 238): “Astrocytes exhibit dynamic changes in second messenger levels and hence gene expression patterns in response to neuronal inputs through neurotransmitter receptors. To date, there are many options for capturing spatiotemporal dynamics of intra- and extra-cellular signalling from the subcellular domain to multicellular levels thanks to the great advance in genetically encoded fluorescent sensors (e.g., GCaMP series for intracellular calcium, intracellular cyclic AMP sensors: Kawata *et al.*, 2022; Massengill *et al.*, 2022, iGluSnFR for extracellular glutamate: Aggarwal *et al.*, 2023, and GPCR-based sensors for extracellular neurotransmitters, neuromodulators, and neuropeptides: Muir *et al.*, 2024; also see: Yu *et al.*, 2020a; Hirrlinger & Nimmerjahn, 2022). Several software packages have been developed to analyse the complex calcium dynamics in astrocytes that are also applicable for other optical probes (Wang *et al.*, 2019; for a review: Hirrlinger & Nimmerjahn, 2022). In addition, activity-dependent cell tagging techniques provide optical and genetic access to astrocyte populations that are functionally linked to specific circuits or animal behaviours (Serra *et al.*, 2022; Williamson *et al.*, 2024).”

- In the Calcium imaging subsection, we have added the description (page 9, line 256): “Two-photon calcium imaging allows subcellular functional mapping of astrocytic microdomains responsive to neuronal inputs *in vivo* (Stobart *et al.*, 2018b; Georgiou *et al.*, 2022), and...”
- Also in the Calcium imaging subsection,

Previous manuscript (page 7, line 212): “~~While fine processes of astrocytes show rapid calcium responses with a few hundred milliseconds delays following neuronal inputs, the astrocytic response has generally slow kinetics (Stobart *et al.*, 2018). Therefore, one~~

~~should bear in mind that the~~ functional activity mapping of astrocyte response by means of calcium activity may involve polysynaptic connections. ~~The use of~~ appropriate inhibitors for related receptors or locally blocking neuronal activity with tetrodotoxin or lateral and feedforward inhibitions with picrotoxin may help in-depth examination, though it is still challenging to completely rule out the possibility.”

Revised manuscript (page 10, line 293): “Calcium imaging is the first option for functional examination of the neuron-to-astrocyte connection, but several potential caveats should be noted. First, functional activity mapping of astrocyte response by means of calcium activity may involve polysynaptic connections considering the slow kinetics of astrocytic response (but see Stobart *et al.*, 2018 for rapid calcium response). Using appropriate inhibitors for related receptors, locally blocking neuronal activity with tetrodotoxin, or lateral or feedforward inhibitions with picrotoxin may help with in-depth examinations, though it is still challenging to completely rule out the possibility. Simultaneous imaging of astrocytic calcium dynamics with neuronal bouton calcium activities (Reynolds *et al.*, 2019) or with neurotransmitter dynamics (Oe *et al.*, 2020) would also help examine the functional monosynaptic connection.”

- We have modified the subsection title:

Previous manuscript (page 9, line 280): “Subregional astrocyte heterogeneity defined by circuitry.”

Revised manuscript (page 13, line 406): “Microcircuits unique to specific astrocyte subpopulation.”

- In the above subsection, we have modified:

1.

Previous manuscript (page 9, line 291): “In the hippocampus, astrocytes can be classified into nine groups based on the transcriptional profiles, and one subpopulation termed “glutamatergic astrocytes” is specialised to release glutamate (De Ceglia *et al.*, 2023). This subpopulation is located across the hippocampus and most densely in the dentate gyrus, and is also found in the visual cortex.”

Revised manuscript (page 13, line 411): “In the hippocampus, astrocytes can be classified into nine groups based on the transcriptional profiles, and one subpopulation termed “glutamatergic astrocytes” is specialised to release glutamate (De Ceglia *et al.*, 2023). This subpopulation is located across the hippocampus and most densely in the dentate gyrus, and astrocyte-specific knockdown of vesicular glutamate transporter 1 impaired

contextual fear memory.”

2. we have added,

Previous manuscript (page 10, line 295): “Currently, if and how the molecularly and functionally defined ...”

Revised manuscript (page 14, line 428): “All these accumulating lines of evidence suggest that the molecularly defined astrocyte subpopulations are linked to the particular circuit alteration and behavioural dysfunction, while if and how the molecularly and functionally defined ...”

We have rephrased the followings for readability.

- In the subsection, Neuron-astrocyte proximity assay:

1.

Previous manuscript (page 3, line 63): “In this approach, a FRET donor, green fluorescent protein (GFP), is expressed on the plasma membrane of astrocytes by fused with the platelet-derived growth factor receptor (PDFGR) transmembrane domain, and an acceptor, mCherry, is fused with the Neurexin-1 transmembrane domain to be located at the presynaptic axon terminals.”

Revised manuscript (page 5, line 122): “In this approach, a FRET donor, green fluorescent protein (GFP) fused with the transmembrane domain of platelet-derived growth factor receptor (PDFGR), and an acceptor, mCherry fused with the transmembrane domain of Neurexin-1, are expressed on the plasma membrane of astrocytes and at the presynaptic axon terminals, respectively.”

2.

Previous manuscript (page 3, line 71): “Without using super-resolution microscopy, NAPA allows evaluation of contact proximity at nanoscale with their locations and densities in genetically specified neuron-astrocyte pairs under both fixed and live conditions.”

Revised manuscript (page 5, line 130): “Without using super-resolution microscopy, NAPA allows the evaluation of locations and densities of nanoscale proximate contacts between genetically specified neuron-astrocyte pairs under fixed and live conditions.”

3.

Previous manuscript (page 4, line 95): “Third, NAPA under two-photon microscopy *in vivo*. Considering that the astrocyte-synapse contacts undergo structural remodeling during

lactation, sleep-wake cycle, and learning, as well as under pathological conditions (Lawal *et al.*, 2022), *in vivo* live NAPA imaging will provide structural dynamics of tripartite synapses under physiological and pathological conditions with circuit identities.”

Revised manuscript (page 6, line 156): “Third, *in vivo* live NAPA imaging under two-photon microscopy will enable chronic tracking of the structural remodelling process of tripartite synapses under physiological and pathological conditions with circuit identities.”

- In the Calcium imaging subsection:

Previous manuscript (page 7, line 196): “When glutamatergic projections from the medial prefrontal cortex (mPFC), amygdala, and ventral hippocampus (vHip) were optogenetically stimulated, respectively, the postsynaptic neuronal response patterns were in line with the stimulated axonal projections. However, ...”

Revised manuscript (page 9, line 277): “When glutamatergic projections from the medial prefrontal cortex (mPFC), amygdala, and ventral hippocampus (vHip) were **each** optogenetically stimulated, the postsynaptic neuronal response patterns **aligned with the respective** stimulated axonal projections. By contrast, ...”

We have also carefully reviewed the manuscript and corrected grammatical errors. Please see the revised manuscript with track changes for the details and other minor modifications.

Dear Professor Inoue,

Re: JP-TR-2025-287485R1 "Recent optical approaches for anatomical and functional dissection of neuron-astrocyte circuitry" by Yoshiki Hatashita and Takafumi Inoue

We are pleased to tell you that your paper has been accepted for publication in The Journal of Physiology.

Authors should note that it is too late at this point to offer corrections prior to proofing. Major corrections at proof stage, such as changes to figures, will be referred to the Editors for approval before they can be incorporated. Only minor changes, such as to style and consistency, should be made at proof stage. Changes that need to be made after proof stage will usually require a formal correction notice.

Yours sincerely,

Laura Bennet
Senior Editor
The Journal of Physiology

P.S. - You can help your research get the attention it deserves! Check out Wiley's free Promotion Guide for best-practice recommendations for promoting your work at www.wileyauthors.com/eoo/guide. You can learn more about Wiley Editing Services which offers professional video, design, and writing services to create shareable video abstracts, infographics, conference posters, lay summaries, and research news stories for your research at www.wileyauthors.com/eoo/promotion.

IMPORTANT NOTICE ABOUT OPEN ACCESS: To assist authors whose funding agencies mandate public access to published research findings sooner than 12 months after publication, The Journal of Physiology allows authors to pay an Open Access (OA) fee to have their papers made freely available immediately on publication.

You can check if your funder or institution has a Wiley Open Access Account here: <https://authorservices.wiley.com/author-resources/Journal-Authors/licensing-and-open-access/open-access/author-compliance-tool.html>.

EDITOR COMMENTS

Reviewing Editor:

Many thanks for the thorough revision of the manuscript - both referees are satisfied with the revised version and have not raised any further points.

REFeree COMMENTS

Referee #1:

The authors addressed all my concerns.

Referee #2:

The authors have addressed the comments to my satisfaction.